# A Randomized Controlled Trial of Fasting and Lifestyle Modification in Patients with Metabolic Syndrome: Effects on Patient-Reported Outcomes

**DOI:** 10.3390/nu14173559

**Published:** 2022-08-29

**Authors:** Michael Jeitler, Romy Lauche, Christoph Hohmann, Kyung-Eun (Anna) Choi, Nadia Schneider, Nico Steckhan, Florian Rathjens, Dennis Anheyer, Anna Paul, Christel von Scheidt, Thomas Ostermann, Elisabeth Schneider, Daniela Koppold-Liebscher, Christian S. Kessler, Gustav Dobos, Andreas Michalsen, Holger Cramer

**Affiliations:** 1Institute of Social Medicine, Epidemiology and Health Economics, Charité—Universitätsmedizin Berlin, Corporate Member of Freie Universität Berlin and Humboldt-Universität zu Berlin, 10117 Berlin, Germany; 2Department of Internal and Integrative Medicine, Immanuel Hospital Berlin, 14109 Berlin, Germany; 3National Centre for Naturopathic Medicine, Southern Cross University, Lismore 2480, Australia; 4Center for Health Services Research, Brandenburg Medical School Theodor Fontane, 16816 Neuruppin, Germany; 5Department of Internal and Integrative Medicine, University of Duisburg-Essen, Evang. Kliniken Essen-Mitte, 45276 Essen, Germany; 6Digital Health Center, Hasso Plattner Institute, University of Potsdam, 14482 Potsdam, Germany; 7Department of Psychology and Psychotherapy, Witten/Herdecke University, 58455 Witten, Germany; 8Institute for General Practice and Interprofessional Care, University Hospital Tuebingen, 72076 Tuebingen, Germany; 9Bosch Health Campus, 70376 Stuttgart, Germany

**Keywords:** fasting, metabolic syndrome, modified DASH diet, Mediterranean diet, lifestyle, relaxation, mindfulness, MICOM (Mind-Body Medicine in Integrative and Complementary Medicine), self-efficacy

## Abstract

Lifestyle interventions can have a positive impact on quality of life and psychological parameters in patients with metabolic syndrome (MetS). In this randomized controlled trial, 145 participants with MetS (62.8% women; 59.7 ± 9.3 years) were randomized to (1) 5-day fasting followed by 10 weeks of lifestyle modification (F + LM; modified DASH diet, exercise, mindfulness; *n* = 73) or (2) 10 weeks of lifestyle modification only (LM; *n* = 72). Outcomes were assessed at weeks 0, 1, 12, and 24, and included quality of life (Short-Form 36 Health Survey Questionnaire, SF-36), anxiety/depression (Hospital Anxiety and Depression Scale, HADS), stress (Cohen Perceived Stress Scale, CPSS), mood (Profile of Mood States, POMS), self-efficacy (General Self-Efficacy Scale, GSE), mindfulness (Mindfulness Attention Awareness Scale, MAAS), and self-compassion (Self-Compassion Scale, SCS). At week 1, POMS depression and fatigue scores were significantly lower in F + LM compared to LM. At week 12, most self-report outcomes improved in both groups—only POMS vigor was significantly higher in F + LM than in LM. Most of the beneficial effects within the groups persisted at week 24. Fasting can induce mood-modulating effects in the short term. LM induced several positive effects on quality of life and psychological parameters in patients with MetS.

## 1. Introduction

The metabolic syndrome (MetS) is a condition characterized by the presence of at least three cardiovascular risk factors, such as abdominal obesity, hypertension, insulin resistance, and dyslipidemia [1]. It affects approximately one in three American adults and one in four adults in Europe [2,3]. Most cardiovascular risk factors can be influenced by patient behavior; this is particularly true for poor diet, lack of exercise, and psychological stress [4].

Epidemiological studies highlight the role of psychological risk factors such as psychosocial stress, depression and, anxiety in patients with MetS and cardiovascular disease [5,6,7]. In addition to physical complaints, patients with MetS often have impaired quality of life and psychological state [8,9].

Cardiovascular behavioral prevention requires comprehensive cardiological programs that consider mind-body interventions in addition to aspects of lifestyle and risk factor management, which should be delivered by interdisciplinary teams of health care professionals [6]. While comprehensive lifestyle modification interventions targeting physical inactivity, diet, and psychosocial stress have shown beneficial effects on cardiovascular risk factors [10,11,12], fewer data are available from these programs on patient-reported outcomes (PRO) like mood, distress, and quality of life [9,13,14].

Mind-body medicine (MBM) is an important method within Complementary and Integrative Medicine (CIM) therapies and is defined as “practices that focus on the interactions among the brain, mind, body, and behaviour” and the ways in which emotional, mental, social, spiritual, and behavioural factors can directly affect heath [15]. More broadly, MBM also includes health-related lifestyle topics to improve patient self-efficacy and self-care, particularly related to exercise, diet, relaxation, and self-care strategies [15].

MBM interventions include structured stress reduction programs, relaxation techniques, meditative movement exercises such as yoga, tai chi, and qigong, and mindfulness, which has been identified as a key method of meditation [16]. It can be described as “a particular way of being characterized by nonjudgmental, moment-to-moment alertness in daily life outside of formal meditation practice” [17]. Various MBM interventions such as yoga, qigong, tai chi, and mindfulness training have been studied in patients with obesity and metabolic syndrome in randomized controlled clinical trials [18,19,20]. In addition to improving cardio-metabolic health, yoga and tai chi in particular have been found to have positive effects on quality of life, psychological well-being, and stress levels in patients with MetS [19,20].

Another CIM intervention that has been very well studied in recent years is fasting [21,22,23,24]. Fasting for a longer period of time, usually from 3 to 21 or more days, has a long tradition in Europe. (Prolonged) fasting can affect the psyche including an induction of mood-modulating effects [25,26,27].

While the aforementioned studies evaluated unimodal interventions in clinical trials, we combined these methods into a comprehensive lifestyle modification program, named MICOM (Mind-Body Medicine in Integrative and Complementary Medicine) [15]. This study aimed to assess effects of fasting followed by the MICOM lifestyle modification intervention in patients with MetS on patient-reported outcomes, compared to a lifestyle modification intervention only.

## 2. Methods

### 2.1. Design

This single-blind, bicenter, randomized controlled study was conducted at the Charité Outpatient Center for Complementary and Integrative Medicine at Immanuel Hospital Berlin, Berlin, Germany, and the Department of Internal and Integrative Medicine, Evang. Kliniken Essen-Mitte, Essen, Germany. The study was approved by the ethics committees of the Charité—Universitätsmedizin Berlin (EA4/141/13) and the University of Duisburg-Essen (14-5733 BO). The study was registered at ClinicalTrials.gov (NCT02099968) prior to patient recruitment. The study was conducted and reported according to the Consolidated Standards of Reporting Trials 2010 guidelines [28]. No changes were made to the study protocol after initiation of the study.

### 2.2. Participants

Patients were recruited via local newspaper advertisements and in the study departments, screened for eligibility by a study nurse, and, if they appeared eligible, assessed by a study physician. If the screened patients met all inclusion criteria and no exclusion criteria, informed consent was obtained, and they were enrolled in the study.

Patients presenting a metabolic syndrome according to the NCEP-ATP-III criteria (National Expert Panel on Detection, Evaluation, and Treatment of High Blood Cholesterol in Adults) were included. They also had to be diagnosed with systolic hypertension and/or additional subclinical atherosclerosis (<50% coronary artery stenosis, <50% carotid artery stenosis, peripheral artery disease stage 1).

Exclusion criteria included (1) diabetes mellitus type 1 or insulin bolus therapy, (2) coronary artery disease, myocardial infarction, pulmonary embolism, or stroke within the past 3 months, (3) heart failure ≥ NYHA stage I, (4) peripheral artery disease ≥ stage 2, (5) chronic kidney disease ≥ stage 2, (6) eating disorder, dementia or psychosis, and (7) other severe internal diseases.

### 2.3. Interventions

Both interventions were group-based and delivered by a multidisciplinary team of health professionals such as nutritional counselors, sport therapists, and mind-body therapists with board certified education. They incorporated aspects of the Mind-Body program which was originally developed by the Benson-Henry Mind/Body Medical Institute at Harvard Medical School and further developed and further developed and adapted to German needs, as described in the MICOM (Mind-Body Medicine in Integrative and Complementary Medicine) program [15]. In this program a focus is set on mindfulness and specific group trainings that are rooted in psychoneuroendocrinology, using formal meditation and gentle yoga exercises. Nutritional education included lectures, cooking workshops, as well as group support.

### 2.4. Fasting and Lifestyle Modification (F + LM)

Participants started with 2 vegan days (max. 1200 kcal/day), followed by 5 days of fasting (max. 350 kcal/day), and a stepwise reintroduction of food. Then they participated in the 10-week MICOM comprehensive multimodal lifestyle modification intervention with weekly 6-h sessions [29]. These included lectures and cooking classes on whole-food vegetarian diets with emphasis on a plant-based Mediterranean diet and a modified DASH diet, intermittent fasting (rice day once a week), and recommendations for certain cardioprotective foods [30,31,32,33]. Furthermore, moderate aerobic exercise such as walking and progressive muscle relaxation, mindfulness meditation, and yoga were trained. After completion of the program, monthly group sessions were offered during weeks 13 to 24 to ensure longer-term adherence.

### 2.5. Lifestyle Modification (LM)

The lifestyle intervention was like the one performed for the F + LM group, but without the initial fasting intervention. The program consisted of 55 h of group intervention over a period of 10 weeks. After the end of the program, in months 13 to 24, monthly group sessions were offered to ensure longer-term adherence.

### 2.6. Randomization

Patients were randomly assigned in a 1:1 ratio to either F + LM or LM by block randomization with randomly varying block lengths, stratified by (a) study center and (b) taking/not taking antihypertensive medication. The randomization list was generated by a biometrician not involved in patient recruitment or assessment using Random Allocation software (version 1.0, Mahmood Saghaei, Isfahan, Iran) [34]. The list was password protected, and no person other than the biometrician had access to it. On this basis, he prepared sealed, sequentially numbered opaque envelopes containing the assignments to the interventions. After obtaining written informed consent and baseline assessment, the study physician opened the envelope with the lowest number to reveal the assignment for that patient.

### 2.7. Outcome Measures

Outcomes were assessed at baseline and at 1, 12 and 24 weeks after randomization by a blinded outcome assessor who was not involved in patient recruitment, allocation, or treatment. Physician-assessed outcomes, laboratory parameters, safety data, and explorative experimental variables (immune function, microbiome) were also assessed and have been reported elsewhere [35]. In the here presented analysis, the following PRO were used and will be reported:Quality of life on the Medical Outcomes Study 36-Item-Short-Form (SF-36) including eight dimensions of health: physical functioning (10 items), social role functioning (2 items), physical role functioning (4 items), emotional role functioning (3 items), mental health (5 items), vitality (4 items), bodily pain (2 items), and general health perceptions (5 items), and the physical and mental component score [36].Anxiety and depression on the Hospital Anxiety and Depression Scale (HADS) [37], a 14-item scale for the assessment of anxiety and depression symptoms.Stress on the Cohen Perceived Stress Scale (CPSS) for the assessment of subjective stress within the last week including 14 items about current levels of experienced and perceived stress [38].Mood on the The Aktuelle Stimmungsskala (ASTS), the German version of the Profile of Mood States (POMS) [39]. The POMS is a 35-item instrument that measures four domains of mood disfunction assessing the subscales vigor, fatigue, depression, and anger [40].Self-efficacy on the 10-item General Self-Efficacy Scale (GSE) [41]. Self-efficacy is a construct that “reflects an optimistic self-belief that one can master a difficult task or cope with adversity” [41]. Self-efficacy makes it easier to set goals, make efforts, persevere in the face of obstacles, and recover from setbacks. It can be considered a resource for resilience.Mindfulness on the 15-item Mindfulness Attention Awareness Scale (MAAS) [42]. Mindfulness is characterized by the following three qualities: (1) intentional, (2) related to the present moment, and (3) nonjudgmental.Self-compassion on the 26-Item Self-Compassion Scale (SCS) [43]. Self-compassion refers to a positive basic attitude toward oneself even in difficult life situations. This personality trait has proven to be an effective protective factor that promotes emotional resilience.

### 2.8. Sample Size Calculation and Statistical Analysis

The sample size was calculated a priori using G*Power software [44]. Based on prior research on multimodal lifestyle interventions [10,45], yoga [46], mindfulness [47], and Mediterranean diet [48], a between-group effect size of d = 0.5 was expected. A level 2.5% t-test requires a total of 64 patients per group to detect a respective group difference with a statistical power of 80%. Accounting for a potential loss of power because of a maximum of 10% dropouts, it was intended to include a minimum of 142 participants in this trial.

All analyses were performed intention-to-treat, which means that all participants were randomized, whether or not they provided a complete data set or adhered to the study protocol. Missing data were multiply imputed by Markov chain Monte Carlo methods [49,50], yielding a total of 50 complete data sets.

Group differences were analyzed by univariate analyses of covariance (ANCOVA) modeling the outcome at week 1, 12, or 24 as a function of the treatment group (classified factor), the stratification factors study center (classified covariate) and baseline antihypertensive medication intake (classified covariate), and the respective baseline value (linear covariate). Afterwards, the 50 estimates of group differences were combined to produce overall effect size estimates, 95% confidence intervals, and p-values. Within-group changes over time were analyzed by paired *t*-tests. *p*-Values ≤ 0.05 were considered significant.

All analyses were performed using the Statistical Package for Social Sciences software (IBM SPSS Statistics for Windows, release 22.0. Armonk, NY, USA: IBM Group).

## 3. Results

### 3.1. Patients

A total of 452 patients were screened by telephone and 258 were excluded because they did not meet the inclusion criteria (Figure 1). A further 49 patients were excluded after medical assessment onsite. Finally, 145 participants were enrolled after informed consent and were randomized to the F + LM group (*n* = 73) or the LM group (*n* = 72). A total of 15 participants each were lost to follow-up in the F + LM and LM groups during the study period (Figure 1).

Patients were recruited between April 2014 (first patient in) and December 2014. The last follow-up assessment was completed in December 2015 (last patient out).

Participants’ characteristics are provided in Table 1. About two thirds of the participants were female and about one third had a university degree (mean age 59.7 ± 9.3 years). Participants had a BMI of 33.3 ± 4.5 kg/m^2^. Participants in the F + LM group attended a mean of 8.2 ± 2.3 (82%) out of 10 possible intervention sessions; those in the LM group attended 7.1 ± 3.5 (71%) sessions (*p* = 0.124).

### 3.2. Outcome Measures

Detailed results are provided in Table 2 and Figure 2. At week 1, POMS depression (Δ = −4.34; 95% confidence interval [CI] = −8.67 to −0.01) and fatigue scores (Δ = −4.91; 95% CI = −8.82 to −1.01) were significantly lower in F + LM compared to LM. From week 0 to 12, most self-report outcomes (i.e., quality of life on SF-36, anxiety/depression on HADS, stress on CPSS, mood on POMS, self-efficacy on GSE, mindfulness on MAAS, and self-compassion on SCS) improved in both groups in the range of small to moderate effect sizes; however, only POMS vigor was significantly higher in F + LM than in LM (Δ = 3.60; 95% CI = 0.30 to 6.90). Most positive within-group effects persisted in week 24; only self-compassion was significantly higher in the LM than in F + LM at this time point (Δ = 2.41; 95% CI = 0.69 to 4.16).

## 4. Discussion

This study investigated the effects of fasting, followed by a multimodal lifestyle modification intervention as described in the MICOM Model, on patient-reported outcomes in patients with MetS compared to a lifestyle modification intervention only. After the multimodal lifestyle modification intervention, we found improvements for all outcome measures, namely quality of life, anxiety, depression, stress, mood as well as self-efficacy, mindfulness, and self-compassion, which persisted at the follow-up after 24 weeks. Comparing the groups (1) at week 1, depression and fatigue scores (POMS) were significantly lower; (2) at week 12, only vigor (POMS) was significantly higher; and (3) at week 24, self-compassion was significantly lower in the group that fasted additionally.

The study participants were demographically heterogeneous. More than 60% were women. The distribution of educational attainment was broad, with a slight shift toward those with higher education. The inclusion criteria of the study led to a large proportion of patients who were German adults with metabolic syndrome. This suggests that the results of the study should be transferable to a large proportion of the German population. Baseline scores indicated the studied population having impaired quality of life and moods as well as depressive symptomatology and anxiety upon study entry.

The five-day fast (max. 350 kcal/day) implemented at week 1 reduced depression and fatigue scores in the F + LM group. This is in line with other findings, as especially (prolonged) fasting is known to influence the psyche positively, inducing mood-modulating effects [25,26,27]. Furthermore, participants in the F + LM group had a higher vigor score at the end of fasting as well as at the 12-week follow-up. Recent studies showed that fasting can increase vigor and vitality, emotional balance, and daytime concentration [26,51]. Fasting is also associated with increased availability of neurotrophic factors in the brain, including serotonin, endogenous opioids, and endocannabinoids, which may also contribute to improved vigor and mood [52,53].

Compared to lifestyle modification only, fasting significantly induced several positive short-term effects in physical parameters, which are reported in detail elsewhere. Especially the observed moderate weight loss and the reduction in waist circumference could lead to positive psychological experiences and may have mood-lifting effects. It should be mentioned that fasting has also been reported to potentially have negative effects on mood, e.g., increased irritability and mood lability [25,27]. However, such side effects were not detectable in our study.

Interestingly, a recent study reported differences in psychological well-being depending on whether or not participants had previous experience with fasting [27]. The group with fasting novices had overall more negative psychological states (i.e., increased appetite, increased negative moods, increased stress, and reduced vitality) during the fast. Therefore, previous fasting experience may buffer negative feelings during a fast. However, previous fasting experiences were not recorded in our study.

We found that self-compassion was significantly higher in participants who received only lifestyle change than in participants who received fasting plus lifestyle change. This finding seems to contradict the hypothesized positive effects of fasting. We would not assume that this finding indicates a harmful effect of fasting. The authors are not aware of any other studies on self-compassion and fasting. However, this finding is interesting and may be worth further investigation.

Several parameters may have clinically relevant effects. The minimal clinically important difference (MCID) for the SF-36 physical component score is defined as a value of 2 and for the SF-36 mental component score as 3 points; in our study the increases were greater than 2 and 3 points, respectively, for both groups, resulting in a relevant clinical improvement [54]. The MCID for HADS in patients with cardiovascular disease is defined at a value of 1.7 points, which the study participants also achieved at the 12-week visit in both groups [55]. No MCID is available for CPSS and POMS, but the effects at the 12- and 24-week follow-ups are also in the range of moderate effect sizes.

Various MBM interventions such as yoga, qigong, tai chi, and mindfulness training have been studied in patients with obesity and metabolic syndrome in randomized controlled clinical trials [18,19,20]. In addition to improving cardio-metabolic health, yoga and tai chi in particular have been shown to improve stress, quality of life, and psychological parameters in patients with MetS [20,56].

The MBM lifestyle modification intervention led to an improvement of self-efficacy, mindfulness, and self-compassion in our study population. These parameters could interact with each other. Further studies to explore explanatory models for the effects of MBM are warranted. In particular, mediating variables on mindfulness, self-efficacy, and compassion, and the effects on physical and psychological parameters, need to be examined more closely.

Our study has several limitations. First, the implementation of a control group with no initial fasting could lead to nonspecific effects due to disappointment effects and may also have influenced the study results. Second, we have not assessed participants’ attitudes and motivation regarding the intervention. Third, the time intensity of the two programs was slightly different, as additional fasting was performed in the F + LM group. Another limitation is the predominant use of generic questionnaires in this study instead of condition-specific ones. On the one hand, this was done because epidemiological studies on metabolic syndrome also predominantly use generic instruments [9], which facilitates the comparison of the study results with those in the population. On the other hand, due to the general lack of specific subjective symptoms in metabolic syndrome, there are hardly any specific questionnaires and, accordingly, previous studies have relied on questionnaires specific to individual symptoms (e.g., obesity). Nevertheless, the lack of specific measures is a limitation, and future studies should use condition-specific questionnaires, at least in addition to generic ones. Finally, the extent to which patients adhered to the program by using stress reduction techniques at home was not assessed.

## 5. Conclusions

In summary, the results of this study point to beneficial and clinically relevant effects of fasting and intensified lifestyle modification on quality of life and psychological parameters. In addition, the used combination of interventions appears to have a sustained effect that needs to be further demonstrated in studies with longer-term follow-up. Further trials could compare comprehensive lifestyle interventions with less intensive and even unimodal interventions and assess cost-effectiveness. Further evaluation of mediating variables such as mindfulness, self-efficacy, and self-compassion in relation to psychological parameters, and interaction with improvements in physical and psychological parameters in the context of complex lifestyle interventions, is warranted.

## Figures and Tables

**Figure 1 nutrients-14-03559-f001:**
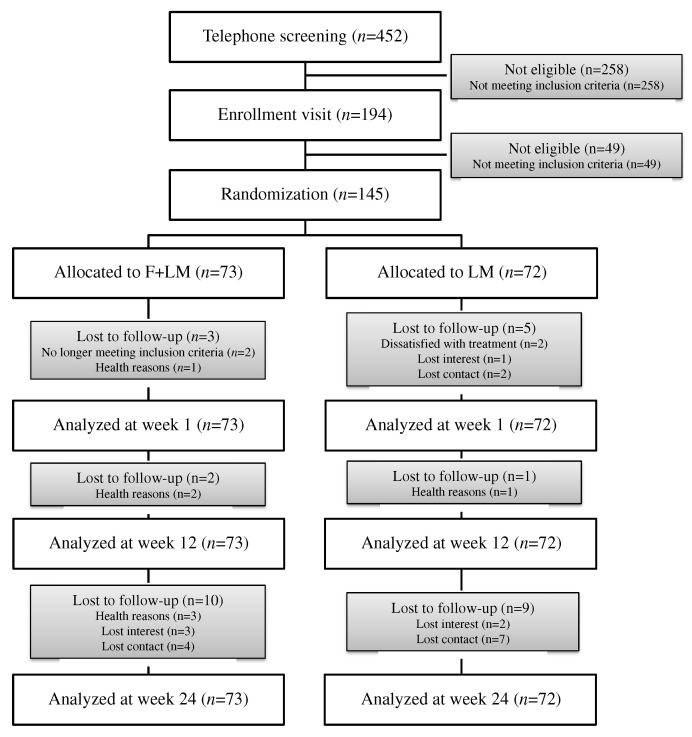
Study flow chart. F + LM, fasting and lifestyle modification; LM, lifestyle modification.

**Figure 2 nutrients-14-03559-f002:**
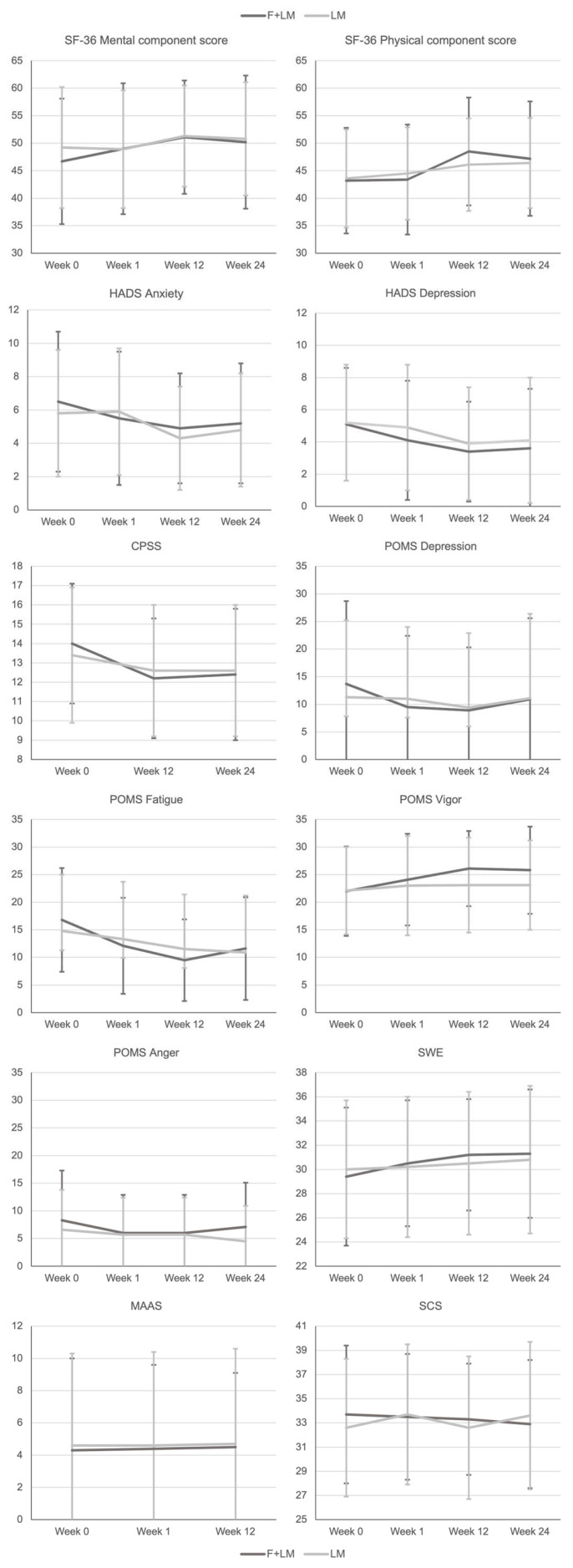
Multiplot on patient-reported outcomes. CPSS, Cohen Perceived Stress Scale; F + LM, fasting and lifestyle modification; LM, lifestyle modification; GSE, General Self-Efficacy Scale; HADS, Hospital Anxiety and Depression Scale; MAAS, Mindfulness Attention Awareness Scale; POMS, Profile of Mood States; SCS, Self-Compassion Scale; SF-36, Short-Form 36 Health Survey Questionnaire.

**Table 1 nutrients-14-03559-t001:** Baseline sociodemographic and clinical characteristics. If not otherwise denoted, values are reported as mean ± standard deviation.

	Total (*n* = 145)	F + LM (*n* = 73)	LM (*n* = 72)
Sociodemographic characteristics			
Gender female n (%)	91 (62.8%)	48 (65.8%)	43 (59.7%)
Age years	59.7 ± 9.3	58.6 ± 10.8	60.8 ± 7.5
Marital status n (%)			
Single	15 (10.3%)	6 (8.2%)	9 (12.5%)
Married	98 (67.6%)	49 (67.1%)	49 (68.1%)
Divorced	21 (14.5%)	12 (16.4%)	9 (12.5%)
Widowed	5 (3.4%)	3 (4.1%)	2 (2.8%)
Education n (%)			
Secondary modern school (“Hauptschule”) qualification	25 (17.2%)	9 (12.3%)	16 (22.2%)
High school (“Realschule”) qualification	41 (28.3%)	19 (26.0%)	22 (30.6%)
A level (“Abitur”)	18 (12.4%)	12 (16.4%)	6 (8.3%)
University degree	53 (36.6%)	28 (38.3%)	25 (34.7%)
Employment n (%)			
Employed full-time	41 (28.3%)	20 (27.4%)	21 (29.2%)
Employed part-time	19 (13.1%)	11 (15.1%)	8 (11.1%)
Occasionally	5 (3.4%)	3 (4.1%)	2 (2.8%)
On sick leave	3 (2.1%)	2 (2.7%)	1 (1.4%)
Unemployed	3 (2.1%)	2 (2.7%)	1 (1.4%)
Retired age-related	44 (30.3%)	20 (27.4%)	24 (33.3%)
Retired health-related	15 (10.3%)	7 (9.6%)	8 (11.1%)
Homekeeper	10 (6.9%)	6 (8.2%)	4 (5.6%)
Clinical characteristics			
Weight (kg)	97.0 ± 15.8	98.1 ± 16.1	95.9 ± 15.5
Body Mass Index (kg/m^2^)	33.3 ± 4.5	33.7 ± 4.5	32.8 ± 4.5
Clinical systolic BP (mmHg)	140.0 ± 16.8	138.9 ± 14.4	141.2 ± 19.0
Clinical diastolic BP (mmHg)	88.9 ± 10.9	88.3 ± 10.6	89.5 ± 11.2
Waist circumference (cm)	113.1 ± 10.8	114.1 ± 10.5	112.1 ± 11.1
HDL cholesterol (mg/dL)	55.0 ± 17.6	53.4 ± 16.0	56.6 ± 19.0
Triglyceride (mg/dL)	181.8 ± 194.0	188.0 ± 210.6	175.5 ± 111.1
Blood glucose (mg/dL)	111.7 ± 20.5	113.3 ± 18.9	110.1 ± 22.0

Abbreviations: F + LM, fasting and lifestyle modification; LM, lifestyle modification.

**Table 2 nutrients-14-03559-t002:** Effects of the study interventions on patient-reported outcomes. Bold *p*-values indicate significant group differences (*p* < 0.05).

		Week 0	Week 1				Week 12				Week 24			
Outcome	Group	Mean ± SD	Mean ± SD	P_within_	P_between_	Cohen’s d	Mean ± SD	P_within_	P_between_	Cohen’s d	Mean ± SD	P_within_	P_between_	Cohen’s d
SF-36 Physical functioning	F + LM	73.3 ± 17.8	75.8 ± 18.8	0.077	0.858	0.03	83.1 ± 17.4	<0.001	0.324	0.15	82.9 ± 16.5	<0.001	0.541	0.10
	LM	74.2 ± 20.6	75.9 ± 21.2	0.118			81.4 ± 19.1	<0.001			81.2 ± 22.5	<0.001		
SF-36 Physical role functioning	F + LM	61.0 ± 40.0	58.1 ± 39.0	0.453	0.244	0.21	78.0 ± 33.3	<0.001	0.104	0.31	69.2 ± 37.5	0.092	0.756	0.06
	LM	63.2 ± 41.1	65.0 ± 38.9	0.595			70.5 ± 36.0	0.059			68.1 ± 38.3	0.147		
SF-36 Bodily pain	F + LM	56.8 ± 22.9	57.3 ± 24.0	0.812	0.119	0.28	69.6 ± 24.0	<0.001	0.238	0.28	64.1 ± 24.5	0.006	0.545	0.13
	LM	62.0 ± 26.8	63.0 ± 25.6	0.654			65.0 ± 29.2	0.337			69.1 ± 27.7	0.008		
SF-36 General health perceptions	F + LM	58.3 ± 16.8	63.5 ± 17.1	0.001	0.238	0.27	70.1 ± 17.7	<0.001	0.370	0.20	69.3 ± 16.8	<0.001	0.527	0.14
	LM	58.1 ± 16.7	59.5 ± 18.4	0.367			65.9 ± 19.0	<0.001			66.1 ± 19.1	<0.001		
SF-36 Vitality	F + LM	55.0 ± 19.7	58.4 ± 19.6	0.014	0.695	0.05	67.7 ± 17.0	<0.001	0.121	0.25	63.5 ± 20.2	<0.001	0.678	0.08
	LM	56.8 ± 21.4	58.4 ± 21.0	0.192			63.5 ± 21.0	<0.001			61.8 ± 23.7	0.004		
SF-36 Social role functioning	F + LM	70.4 ± 27.8	75.9 ± 27.7	0.023	0.722	0.07	83.7 ± 23.2	<0.001	0.176	0.25	81.0 ± 24.0	<0.001	0.922	0.02
	LM	80.7 ± 23.4	79.4 ± 25.3	0.514			82.2 ± 24.6	0.493			83.3 ± 21.0	0.275		
SF-36 Emotional role functioning	F + LM	68.9 ± 37.8	70.9 ± 39.1	0.605	0.495	0.13	77.3 ± 31.9	0.030	0.504	0.12	77.1 ± 33.7	0.012	0.957	0.01
	LM	72.2 ± 39.6	71.1 ± 39.6	0.722			80.4 ± 34.4	0.036			80.5 ± 34.2	0.017		
SF-36 Mental health	F + LM	66.9 ± 18.8	71.5 ± 19.5	0.002	0.444	0.12	75.6 ± 16.8	<0.001	0.128	0.23	73.0 ± 20.2	<0.001	0.887	0.03
	LM	70.7 ± 17.8	71.4 ± 18.2	0.536			75.5 ± 15.2	0.001			74.0 ± 18.2	0.027		
SF-36 Physical component score	F + LM	43.2 ± 8.9	43.4 ± 8.4	0.739	0.131	0.24	48.5 ± 8.4	<0.001	0.177	0.26	47.2 ± 8.2	<0.001	0.990	0.01
	LM	43.6 ± 9.6	44.5 ± 10.0	0.223			46.1 ± 9.8	0.006			46.4 ± 10.4	0.002		
SF-36 Mental component score	F + LM	46.7 ± 11.4	49.0 ± 11.9	0.005	0.129	0.22	51.1 ± 10.3	<0.001	0.096	0.24	50.2 ± 12.1	<0.001	0.594	0.10
	LM	49.2 ± 11.0	48.9 ± 10.7	0.580			51.3 ± 9.2	0.017			50.8 ± 10.3	0.065		
HADS Anxiety	F + LM	6.5 ± 4.2	5.5 ± 4.0	<0.001	0.303	0.16	4.9 ± 3.3	<0.001	0.227	0.18	5.2 ± 3.6	<0.001	0.206	0.21
	LM	5.8 ± 3.8	5.9 ± 3.8	0.751			4.3 ± 3.1	<0.001			4.8 ± 3.4	0.004		
HADS Depression	F + LM	5.1 ± 3.5	4.1 ± 3.7	<0.001	0.386	0.15	3.4 ± 3.1	<0.001	0.432	0.13	3.6 ± 3.7	<0.001	0.740	0.07
	LM	5.2 ± 3.6	4.9 ± 3.9	0.367			3.9 ± 3.5	<0.001			4.1 ± 3.9	0.004		
CPSS	F + LM	14.0 ± 3.1	NA	NA	NA	NA	12.2 ± 3.1	<0.001	0.070	0.33	12.4 ± 3.4	<0.001	0.064	0.37
	LM	13.4 ± 3.5	NA	NA			12.6 ± 3.4	0.005			12.6 ± 3.4	0.020		
POMS Depression	F + LM	13.7 ± 15.0	9.5 ± 12.9	<0.001	0.049	0.30	8.9 ± 11.4	<0.001	0.397	0.11	10.9 ± 14.7	0.010	0.504	0.12
	LM	11.3 ± 13.9	11.0 ± 13.0	0.747			9.4 ± 13.5	0.065			11.1 ± 15.3	0.856		
POMS Fatigue	F + LM	16.8 ± 9.4	12.1 ± 8.7	<0.001	**0.014**	0.50	9.5 ± 7.4	<0.001	0.206	0.23	11.6 ± 9.3	<0.001	0.549	0.12
	LM	14.8 ± 10.2	13.3 ± 10.4	0.096			11.5 ± 9.9	0.001			10.9 ± 10.3	<0.001		
POMS Vigor	F + LM	22.0 ± 8.1	24.1 ± 8.3	0.010	0.629	0.10	26.1 ± 6.8	<0.001	**0.033**	0.45	25.8 ± 7.9	<0.001	0.147	0.30
	LM	22.1 ± 7.9	23.0 ± 9.0	0.264			23.1 ± 8.6	0.278			23.1 ± 8.1	0.260		
POMS Anger	F + LM	8.3 ± 9.0	6.0 ± 6.9	0.001	0.089	0.27	6.0 ± 6.9	0.001	0.412	0.14	7.1 ± 8.0	0.087	0.192	0.23
	LM	6.6 ± 7.2	5.7 ± 6.7	0.138			5.7 ± 6.7	0.279			4.5 ± 6.4	0.010		
GSE	F + LM	29.4 ± 5.7	30.5 ± 5.2	0.020	0.244	0.17	31.2 ± 4.6	<0.001	0.206	0.19	31.3 ± 5.3	<0.001	0.262	0.19
	LM	30.0 ± 5.7	30.2 ± 5.8	0.553			30.5 ± 5.9	0.170			30.8 ± 6.1	0.061		
MAAS	F + LM	4.3 ± 0.9	NA	NA	NA	NA	4.4 ± 0.8	0.030	0.696	0.06	4.5 ± 0.8	0.001	0.265	0.18
	LM	4.6 ± 0.7	NA	NA			4.6 ± 0.7	0.183			4.7 ± 0.7	0.169		
SCS	F + LM	33.7 ± 5.5	33.5 ± 4.9	0.702	0.618	0.09	33.3 ± 4.5	0.503	0.698	0.06	32.9 ± 5.0	0.060	**0.007**	0.41
	LM	32.6 ± 6.2	33.7 ± 4.5	0.110			32.6 ± 5.0	0.978			33.6 ± 4.6	0.057		

Abbreviations: CPSS, Cohen Perceived Stress Scale; F + LM, fasting and lifestyle modification; LM, lifestyle modification; GSE, General Self-Efficacy Scale; HADS, Hospital Anxiety and Depression Scale; MAAS, Mindfulness Attention Awareness Scale; NA, not assessed; P_between_, between-group differences (analysis of covariances); P_within_, within-group changes from baseline (paired *t*-test); POMS, Profile of Mood States; SCS, Self-Compassion Scale; SD, standard deviation; SF-36, Short-Form 36 Health Survey Questionnaire.

## Data Availability

Data available on request.

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
