# Peer review of "A Randomized Controlled Trial of Fasting and Lifestyle Modification in Patients with Metabolic Syndrome: Effects on Patient-Reported Outcomes"

_nutrients, 2022, doi:10.3390/nu14173559_

Round 1
Reviewer 1 Report
Jeitler et al. surveyed the effects of fasting, followed by a multimodal lifestyle modification intervention on patient-reported outcomes in patients with metabolic syndrome (compared to a lifestyle modification intervention only). The paper is within the scope of the journal and the topic is interesting and relevant for the field. The research is well described, even if an extensive revision of the text editing is necessary. Indeed, there are many text formatting errors. However, tables and figures are appropriate. Furthermore, the manuscript is well organized, readily understandable and the research of information has been accurate, with updated references.
Author Response
Jeitler et al. surveyed the effects of fasting, followed by a multimodal lifestyle modification intervention on patient-reported outcomes in patients with metabolic syndrome (compared to a lifestyle modification intervention only). The paper is within the scope of the journal and the topic is interesting and relevant for the field. The research is well described, even if an extensive revision of the text editing is necessary. Indeed, there are many text formatting errors. However, tables and figures are appropriate. Furthermore, the manuscript is well organized, readily understandable and the research of information has been accurate, with updated references.
Response: Thank you, we have thoroughly revised the manuscript.
Reviewer 2 Report
The authors conducted a randomized control trial to examine the effects of fasting and lifestyle modification on patient-reported outcomes in patients with metabolic syndrome. A total of 145 patients with metabolic syndrome were randomized to receive fasting plus lifestyle modification or lifestyle modification alone. The authors showed that those receiving fasting plus lifestyle modification demonstrated less fatigue, less depressive mood, and more vigor than those receiving lifestyle modification alone. Given the growing burden of metabolic disorders, the findings of this study are important and could have implications in future research directions.
There are some comments.
Comments:
1. Results: Please describe the dates defining the recruitment period and follow-up period.
2. Results (Table 1): Baseline sociodemographic and clinical characteristics are shown in Table 1. This study focused on the patient-reported outcomes in patients with metabolic syndrome. I would suggest additionally presenting metabolic syndrome components (blood pressure, waist circumference, HDL cholesterol, triglyceride, glucose), if available, and the baseline patient-reported outcomes in this table. These data would allow the readers to understand better the metabolic disorders severity range these study populations had.
3. Results (Table 1): Baseline sociodemographic and clinical characteristics are shown in Table 1. However, whether the two intervention groups differ in these characteristics is unclear. I would suggest conducting statistical tests for the differences between the two intervention groups and presenting the results.
4. Discussion: In the first paragraph, the authors described that “self-compassion was significantly higher in the group that fasted additionally.” However, as shown in Table 2, self-compassion was significantly higher in those receiving lifestyle modification alone than in those receiving fasting plus lifestyle modification. A correction is suggested.
5. Discussion: The authors observed that self-compassion was significantly higher in those receiving lifestyle modification alone than in those receiving fasting plus lifestyle modification. This finding seems to contradict the assumed beneficial effects of fasting. Did this finding suggest a harmful effect of fasting? I would recommend a discussion of this interesting finding.
6. Discussion: In this study, patient-reported outcomes were measured using SF-36, HADS, CPSS, ASTS, GSE, MAAS, and SCS. Most of these, for instance, SF-36, are generic measures. Although generic measures, in general, are useful in comparing across diseases, health conditions, or populations, metabolic syndrome-specific measures (for instance, quality of life) might be particularly useful in detecting change. Conducting research applying such disease-specific patient-reported outcomes measures to examine further the fasting’s health effects in patients with metabolic syndrome would be worthwhile in the future. A discussion is recommended.
Author Response
The authors conducted a randomized control trial to examine the effects of fasting and lifestyle modification on patient-reported outcomes in patients with metabolic syndrome. A total of 145 patients with metabolic syndrome were randomized to receive fasting plus lifestyle modification or lifestyle modification alone. The authors showed that those receiving fasting plus lifestyle modification demonstrated less fatigue, less depressive mood, and more vigor than those receiving lifestyle modification alone. Given the growing burden of metabolic disorders, the findings of this study are important and could have implications in future research directions.
There are some comments.
Comments:
- Results: Please describe the dates defining the recruitment period and follow-up period
Response: We added the follow sentences: “Patients were recruited between April 2014 (first patient in) and December 2014. The last follow-up assessment was completed in December 2015 (last patient out).”
- Results (Table 1): Baseline sociodemographic and clinical characteristics are shown in Table 1. This study focused on the patient-reported outcomes in patients with metabolic syndrome. I would suggest additionally presenting metabolic syndrome components (blood pressure, waist circumference, HDL cholesterol, triglyceride, glucose), if available, and the baseline patient-reported outcomes in this table. These data would allow the readers to understand better the metabolic disorders severity range these study populations had.
Response: We added the metabolic syndrome components as suggested.
|
Total (n=145) |
F+LM (n=73) |
LM (n=72) |
Clinical systolic BP mmHg |
140.0 ± 16.8 |
138.9 ± 14.4 |
141.2 ± 19.0 |
Clinical diastolic BP mmHg |
88.9 ± 10.9 |
88.3 ± 10.6 |
89.5 ± 11.2 |
Waist circumference cm |
113.1 ± 10.8 |
114.1 ± 10.5 |
112.1 ± 11.1 |
HDL cholesterol mg/dl |
55.0 ± 17.6 |
53.4 ± 16.0 |
56.6 ± 19.0 |
Triglyceride mg/dl |
181.8 ± 194.0 |
188.0 ± 210.6 |
175.5 ± 111.1 |
Blood glucose mg/dl |
111.7 ± 20.5 |
113.3 ± 18.9 |
110.1 ± 22.0 |
- Results (Table 1):Baseline sociodemographic and clinical characteristics are shown in Table 1. However, whether the two intervention groups differ in these characteristics is unclear. I would suggest conducting statistical tests for the differences between the two intervention groups and presenting the results
Response: The CONSORT statement explicitly discourages inferential statistical tests for baseline differences between groups: “Tests of baseline differences are not necessarily wrong, just illogical. Such hypothesis testing is superfluous and can mislead investigators and their readers.” Given that we aimed to follow the CONSORT statement, such tests were not conducted. We hope that this is acceptable to the reviewer.
4. Discussion: In the first paragraph, the authors described that “self-compassion was significantly higher in the group that fasted additionally.” However, as shown in Table 2, self-compassion was significantly higher in those receiving lifestyle modification alone than in those receiving fasting plus lifestyle modification. A correction is suggested.
Response: Thank you, we have corrected that.
- Discussion: The authors observed that self-compassion was significantly higher in those receiving lifestyle modification alone than in those receiving fasting plus lifestyle modification. This finding seems to contradict the assumed beneficial effects of fasting. Did this finding suggest a harmful effect of fasting? I would recommend a discussion of this interesting finding
Response: We added following paragraph in the discussion section: “We found that self-compassion was significantly higher in participants who received only lifestyle change than in participants who received fasting plus lifestyle change. This finding seems to contradict the hypothesized positive effects of fasting. We would not assume that this finding indicates a harmful effect of fasting. The authors are not aware of any other studies on self-compassion and fasting. However, this finding is interesting and may be worth further investigation.”
- Discussion: In this study, patient-reported outcomes were measured using SF-36, HADS, CPSS, ASTS, GSE, MAAS, and SCS. Most of these, for instance, SF-36, are generic measures. Although generic measures, in general, are useful in comparing across diseases, health conditions, or populations, metabolic syndrome-specific measures (for instance, quality of life) might be particularly useful in detecting change. Conducting research applying such disease-specific patient-reported outcomes measures to examine further the fasting’s health effects in patients with metabolic syndrome would be worthwhile in the future. A discussion is recommended
Response: We agree and added to the limitation section following paragraph: “Another limitation is the predominant use of generic questionnaires in this study instead of condition-specific ones. On the one hand, this was done because epidemiological studies on metabolic syndrome also predominantly use generic instruments (Saboya PP, Bodanese LC, Zimmermann PR, Gustavo AD, Assumpção CM, Londero F. Metabolic syndrome and quality of life: a systematic review. Rev Lat Am Enfermagem. 2016;24:e2848.), which facilitates the comparison of the study results with those in the population. On the other hand, due to the largely lack of specific subjective symptoms in metabolic syndrome, there are hardly any specific questionnaires and, accordingly, previous studies have relied on questionnaires specific to individual symptoms (e.g., obesity). Nevertheless, the lack of specific measures is a limitation, and future studies should use condition-specific questionnaires at least in addition to generic ones.”